

# Constructing physical-based rainfall landslides prediction model: Insights from rainfall threshold curves database of slope units

Kai Wang[1*], Linmao Xie[1], Shuailong Xie[1], Shaojie Zhang[2*], Yongyang Jiang[3], Ji Zhang[4], Lin Zhu[1], Zhiliu Wang[1], Fuzhou Qi[1]

*1. School of architecture and civil engineering, Zhongyuan University of Technology, Zhengzhou, 450007, China*

*2. Key Laboratory of Mountain Hazards and Earth Surface Process, Institute of Mountain Hazards and Environment, Chinese Academy of Sciences, Chengdu, 610041, China*

*3. Zhejiang Zhongnan Construction Group Steel Structure Co., Ltd, Hangzhou, 311400, China*

*4. Sichuan Institution of Geological Engineering Investigation Group Co.LTD, Chengdu, 610041, China*

**Abstract:** The commonly used rainfall threshold warning method relies heavily on historical rainfall and landslide inventory data, which limits its applicability in regions that lack these data. While physical methods do not rely on landslide inventories to establish warning criteria, the calculation of the safety factor typically requires considerable time. To address these issues, this study integrates physical methods, rainfall threshold warning methods, and slope units to develop a rapid forecasting model for rainfall landslides at a regional scale. A hydrological analysis technique for slope units based on grid cells was developed to calculate the instability probability of slope units. Then, each slope unit was analyzed under 20 levels of antecedent effective precipitation and nearly 200 combinations of rainfall intensity (I) and duration (D) to derive the key fitting parameters α and β of the I-D curves under various rainfall scenarios. The application results from Fengjie County indicate that the model runs in less than 12 min, with missing alarm and false alarm rates of 11.8% and 21.1%, respectively, highlighting its excellent potential for practical application. This study is expected to provide insights for the rapid forecasting of rainfall landslides in the impoverished mountainous regions of developing countries.

**Key words:** Landslide forecasting model, Slope unit, Fitting parameters, Warning database

* Corresponding Authors 1: Kai Wang
E-mail: 6696@zut.edu.cn
* Corresponding Authors 2: Shaojie Zhang
E-mail: sj-zhang@imde.ac.cn



## 1    Introduction

Rainfall-induced landslides at a regional scale are among the most common types of natural hazards worldwide. Reports indicate that in the United States, rainfall-triggered landslides and secondary hazards result in 25–50 fatalities and economic losses of approximately $2 billion annually (He et al., 2016). This loss is even more severe in developing countries in the Third World (Wang et al., 2024; Wang et al., 2021; Wang et al., 2023). In recent years, numerous studies have indicated that regional landslide forecasting is highly effective for hazard prevention and mitigation. Researchers have developed various rainfall landslide forecasting models based on statistical and physical methods (Aristizábal et al., 2016; Baum et al., 2008; Bezak et al., 2016; Bogaard et al., 2018; Cuomo et al., 2021; Liang et al., 2021; Medina et al., 2021; Pinho et al., 2022; Tufano et al., 2021; Wang et al., 2013; Zhang et al., 2021; Zhang et al., 2019). However, there are still several unresolved issues in regional landslide forecasting, making accurate and efficient warnings a significant global challenge.

The first major issue is the selection of forecasting methods. The presented statistical approaches generally depend on historical precipitation and landslide inventory data to construct the rainfall threshold curves. Recently, researchers proposed different types of rainfall threshold curves, including I-D, E-D, E-I, IR-AER, I-P, and I-D-MEAR(Brunetti et al., 2010; Hong et al., 2005; Rosi et al., 2020; Zhuang et al., 2014). The I-D curve is the most extensively used among these types. The I-D curve is typically fitted in either Cartesian coordinates or a double-logarithmic coordinate system, and the equation of the curve is governed by two key fitting parameters, α and β, expressed as follows:

$$I = \alpha D^{\beta} \tag{1}$$

where α and β are derived from the statistical analysis of historical rainfall and landslide data.

Studies indicate that statistical methods are applicable in regions with abundant historical records of rainfall landslides because these areas can provide sufficient samples for fitting the I-D curve(Bezak et al., 2016; Hong et al., 2017; Kanungo et al., 2014; Kim et al., 2020; Ma et al., 2015; Marra, 2018; Pradhan et al., 2018). However, in the poor mountainous regions of the Third World, many areas that are severely affected by landslides lack professional monitoring devices and rain gauges, potentially limiting the application of statistical approaches(Zhang et al., 2021; Zhang et al., 2019). In contrast, physical methods typically rely on hydrological and mechanical analyses to



calculate the safety factors of landslides under different rainfall scenarios, thereby reducing the
reliance on historical rainfall and landslide observation data. In regions where landslide inventory
data are scarce, physical methods could serve as promising alternatives (Zhang et al., 2021; Zhang
et al., 2019). However, physical methods require historical landslide data to validate the accuracy
of the forecasting results, and the safety factor calculation process typically requires a considerable
amount of time. This computational burden increases substantially when considering the stability
analysis of thousands of slopes at the regional scale, making it difficult to ensure the efficiency of
real-time warnings (Zhang et al., 2021).

9         The second issue pertains to the selection of prediction unit. Clearly defined prediction units

enable residents to identify the specific locations where landslides are likely to occur while also
providing guidance for local governments to develop emergency schemes. However, the I-D
warning curves derived from statistical methods can only provide general trends of hazards within
the region but cannot pinpoint the specific locations of landslide occurrences. Grid cells improve
the clarity of the prediction results to some extent, as the specific locations of each grid within the
area are well defined (Zhang et al., 2021). Researchers have employed grid cells to establish multiple
physical forecasting models such as SHALSTAB (Montgomery et al., 1994), SINMAP(Tarboton et
al., 1970), GEOtop-FS(Rigon et al., 2006), Trigrs(Baum et al., 2008), HIRESSS (Rossi et al., 2013),
H-slider(Liang et al., 2021), SHIA_Landslide (Aristizábal et al., 2016), SLIP(Montrasio et al., 2016),
and FSLAM(Guo et al., 2022). However, the morphology of grid cells does not accurately
characterize the topographical features of natural hillslopes (Domènech et al., 2019; Zhang et al.,
2021), resulting in a lack of clear geomorphological significance. In practical applications, a natural
slope can be segmented into a series of grid cells, in which each grid is assigned a different alert
level. This indicates that a high warning level in a grid cell does not mean that the entire slope will
experience a slide.

25         In contrast, slope units can represent the topographical features of landslides more accurately,

and their boundaries are easily discernible in field environments. Currently, there are various
methods for extracting slope units, including the DEM-based hydrological process analysis
method(Turel et al., 2011), r.slopeunits method(Alvioli et al., 2020), curvature watershed
methods(Yan et al., 2021), MIA-HSU methods(Wang et al., 2019), and multi-scale image
segmentation methods (Huang et al., 2021). In recent years, researchers have developed forecasting



models utilizing slope units, validating their promising application potential in predicting rainfall-
induced landslides (Wang et al., 2023; Zhang et al., 2021).
Addressing the issues outlined in regional landslide forecasting, this study focuses on the
integration of slope units, physical methods, and rainfall parameterized warning techniques to
develop a rapid forecasting model applicable to large areas on a scale of thousands of square
kilometers. Within this model, we no longer pay attention to the positional relationship between the
rainfall data of a landslide and the I-D curve, but concentrate on the key fitting parameters $\alpha$ and $\beta$
of the I-D curve for each slope unit. To facilitate this, we developed a rainfall infiltration simulation
technique rooted in grid cells within slope units and subsequently utilized physical methods to
analyze the instability probability for slope units under different rainfall scenarios. For each slope
unit, we designed rainfall scenarios comprising various antecedent rainfall levels combined with
hundreds of rainfall intensity and duration combinations. This allowed us to obtain the key
parameters $\alpha$ and $\beta$ of the I-D curves for different rainfall scenarios, thereby constructing a database
of parameters $\alpha$ and $\beta$ under various antecedent precipitation levels. A case study in Fengjie County,
in the Three Gorges Reservoir area, was conducted to validate the reliability of the proposed method.
This research is expected to provide valuable insights for regional landslide forecasting in
impoverished mountainous areas in the developing world.
## 2   Methodology
### 2.1 The slope unit extraction method MIA-HSU
In this study, we employed the MIA-HSU method to extract slope units(Wang et al., 2021;
Wang et al., 2019; Wang et al., 2023). In the MIA-HSU method, each HSU(homogeneous slope unit)
is defined as a continuous and homogeneous geomorphological entity. This definition implies that
terrain features related to slope and aspect are uniform within each HSU, with boundaries indicating
transitions in topographical features. The MIA-HSU method consists of two steps. The first step
involves partitioning the Digital Elevation Model (DEM) into small regions with homogeneous
terrain characteristics. In this step, the MIA-HSU method utilizes terrain curvature analysis to
identify ridge and valley regions (Figure 1a) and then extracts the morphological skeleton lines of
ridge and valley areas to characterize topographic relief. Morphological algorithms (such as dilation
and erosion) were used to extract the morphological skeletons of ridges, valleys, and flat areas from
the DEM (Figure 1b), ultimately connecting these skeletons into a closed network (Figure 1c). Thus,



each small region within the network exhibits uniform geomorphological characteristics. The
second step involves merging small adjacent regions. The MIA-HSU method employs the principal
component analysis (PCA) method to derive fitted planes from localized terrain regions, followed
by the implementation of vector similarity criteria to merge adjacent small regions, thereby
generating HSUs(Figure 1d).

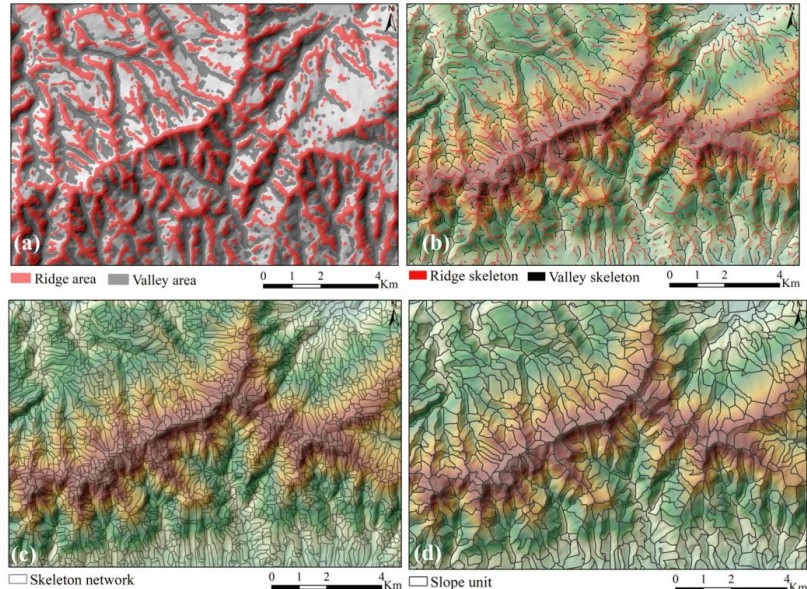

Figure 1 HSU extraction process: a. the identification of ridge and valley areas; b. the morphological skeleton line
extraction for ridge and valley areas; c. the morphological skeleton closed network; d. HSU extraction result
**2.2 The HSU hydrologic simulation technique based on grid cells**
(1)**The identification for row and column information of grid cell within HSUs**
From a geometric perspective, an HSU can be regarded as a spatial polygon that signifies a
landform entity with homogeneous terrain features in the field environment. At the regional scale,
there is obvious heterogeneity in the topography and boundary characteristics among different
HSUs(Wang et al., 2021; Wang et al., 2019; Wang et al., 2023), resulting in the immaturity of
hydrological analysis methods based on slope units. In contrast, hydrological analysis methods
based on grid cells are well-developed. Some researchers have employed grid cells integrated with
an infinite slope model or the limit equilibrium method to conduct regional landslide assessment or
prediction(Gu et al., 2014; Wang et al., 2023; Zhang et al., 2021; Zhuang et al., 2016). In this study,
each HSU was conceptualized as a composition of grid cells with similar microtopographic features,
as illustrated in Figure 2a. For each HSU, we utilized GIS spatial analysis tools to quantify the



number of grid cells contained within it and their corresponding row and column positional
information, thus establishing a comprehensive database that includes the position information of
the grid cells within each HSU.

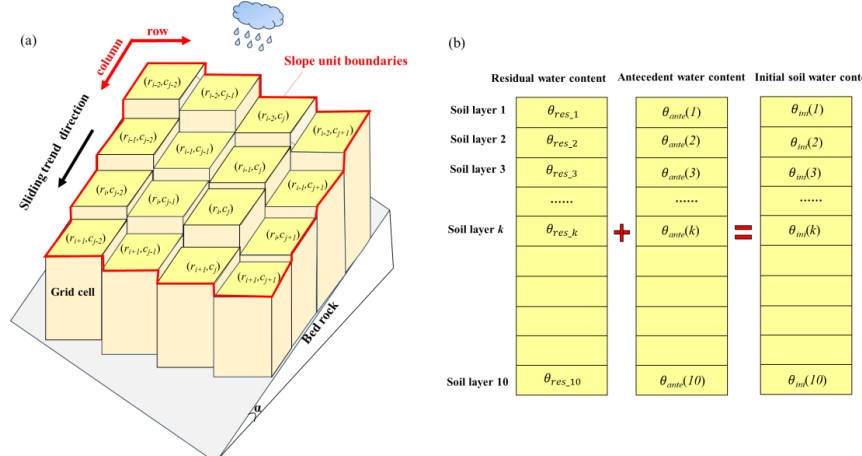

Figure 2 The diagram for HSU-grid cell hydrological connection: a. Grid cells contained within HSU ($r_i$, $c_j$ represent
the row and column of grid cells contained within HSU, respectively)
**(2)Initial water content assignment of HSUs**

8   After obtaining the grid cell information contained within each HSU, conducting a rainfall

infiltration analysis for these grid cells represents a complex and important task. One issue that
cannot be overlooked is initial moisture content. For landslides in the Three Gorges Reservoir area
of China, the soil typically experiences a prolonged dry winter before the rainy season (May to
September). Previous research indicates that the residual moisture content of slopes before the rainy
season averages approximately 7% to 8% (Wang et al., 2023). Accordingly, this study categorizes
the initial water content into two components: the residual moisture content ($\theta_{res}$) and the moisture
content increment caused by antecedent precipitation($\theta_{\text{ante}}$). Here, $\theta_{res}$ reflects the average moisture
level of the soil prior to the rainy season, while $\theta_{\text{ante}}$ indicate the increase in moisture content due to
antecedent effective precipitation prior to landslide occurrence.

18   In this study, each grid cell is stratified into ten soil layers, each with a thickness of 0.2 m

(Figure 2b). For the Three Gorges Reservoir area, the regional landslides triggered by rainfall were
mainly shallow (with thicknesses of 2-3 m). Therefore, variations in residual moisture content
within the soil depth were disregarded, and the same residual moisture content value was assigned
to each soil layer. Following this, we applied steady-state infiltration theory to simulate the



distribution of moisture content across the soil layers influenced by antecedent precipitation, thereby

allocating the antecedent rainfall to each soil layer. The calculation for $\theta_{ini}$ of each soil layer within

the grid cell is as follows:

$$\theta_{\text{ini}}(k)= \theta_{ante}(k) +\theta_{res} \quad (k=1,2,3...n) \tag{2}$$

Where $n$ represents the number of soil layers, and here $n = 10$; $\theta_{\text{ini}}(k)$ indicates the initial

moisture content of each soil layer; $\theta_{\text{ante}}(k)$ refers to the moisture change in each soil layer due to

previous precipitation; $\theta_{\text{res}}$ stands for the residual moisture content in the grid cell.

**(3)Rainfall infiltration process simulation of grid cell**

After obtaining the initial moisture content distribution, the 1-dimensional Richards infiltration

equation was used to solve the moisture content distribution in the grid cell during the rainfall

infiltration process.

$$\frac{\partial \theta}{\partial t} = \frac{\partial}{\partial z}[D(\theta).\frac{\partial \theta}{\partial z}] - \frac{\partial \mathrm{K}(\theta)}{\partial z} \tag{3}$$

Where $D(\theta)$ represents the value of soil water diffusivity under unsaturated conditions and has

$D(\theta) = K(\theta)/\dfrac{d\theta}{d\psi_m}$ .

The finite difference scheme outlined above was formulated for numerical simulation of

hydrological processes. The lower boundary, identified as impermeable, is based on the maximum

soil depth of the grid cell. The upper boundary of each grid cell was designated as an infiltration

boundary. When the rainfall intensity $I(t)$ is less than the infiltration capacity of the topsoil, all

precipitation infiltrates into the soil, and no runoff is generated. In this scenario, the infiltration

boundary of precipitation was governed by the following differential equation:

$$-D(\theta)\frac{\partial \theta}{\partial z} + K(\theta) = I(t) \tag{4}$$

When the rainfall intensity exceeded the soil infiltration capacity, the excess portion was

transformed into overland flow. At this point, the rainfall infiltration boundary was governed by the

following equation:

$$\theta=\theta_s \tag{5}$$

Where $\theta_s$ is the saturated water content of the grid cell.

**(4)Soil water content generation of HSU**

Following the calculation of the soil moisture for individual grid cells, the soil water

distribution of the HSU was computed as follows:



$$\theta_{HSU}(k) = \frac{\sum_{k=1}^{n} \theta(k)}{n} \qquad (6)$$

where $\theta_{\mathrm{HSU}}(k)$ represents the moisture content of the kth layer of the HSU, $\theta(k)$ denotes the
moisture content of the kth layer in the grid cell. $n$ is the number of soil layers ($n = 10$).
*2.3 HSU$_{prob}$*:**the calculation of instability probability of HSUs**
(1) Profile extraction
After calculating the soil water content within each HSU, analyzing the stability of HSUs
during the rainfall infiltration process is another important task. At present, the time required to
carry out 3D analysis for each HSU on a large regional scale is too large, so extracting the calculation
profile of the HSU becomes a reasonable selection. Currently, there is no uniform method for
extracting the calculation profile of HSUs. Some reasonable assumptions are summarized as follows:
the position of the profile line should reflect the elevation difference between the front and back
edges of the slope, and the centroid point of the HSUs should be on the calculated profile to ensure
that the soil weight on both sides of the calculated section is relatively uniform, and the areas of the
two sections should be close to each other.
Based on these considerations, we developed a fast extraction algorithm HSU-profil (Wang et
al., 2021; Wang et al., 2023) for HSU profiles at large regional scales, which can be divided into
three steps:
First, the highest elevation point $H$ of the HSU polygon is connected to centroid point C to
obtain line segment $L_1$, which intersects the HSU polygon at point $J_1$ (Figure 3b). Line segment $L_1$
divides the HSU polygon into two parts, and the areas of the two parts, $S_1$ and $S_2$ are calculated    to
obtain the area ratio A= $S_1$/ $S_2$.
Next, the lowest elevation point $L$ and centroid $C$ are connected to form line segment $L_2$, as
shown in Figure 3 b. Determine the intersection point $J_2$ between $L_2$ and the polygon of the slope
unit is determined. At this point, the HSU was divided into two parts by line segment $L_2$, and the
areas of the two parts, $S_3$ and $S_4$, were calculated to obtain the area ratio $B=S_3$/ $S_4$.
Finally, |A| and |B| are compared. A smaller absolute value of A indicates that line segment $L_1$
divides the areas on both sides of the HSU polygon more evenly. In this case, $L_1$ is selected as the
profile line. Otherwise, the line segment $L_2$ was chosen as the profile line.
(2) Calculation of safety factor $Fs$ calculation



For each HSU, the Monte Carlo method was used to generate a large number of potential
polyline-type slip surfaces (Figure 3c), and the random walk method(Greco, 1996) was employed
to search for the critical slip surface. The infinite slope model was used to calculate the safety factor
Fs of each potential slip surface as follows:

$$F_s = \frac{\tan\varphi}{\tan\alpha} + \frac{c + u_s \tan(\varphi^b)}{\gamma_s D_s \cos\alpha \sin\alpha}$$
(7)

where $c$ is the effective cohesion of the soil, $\varphi$ is the effective internal friction angle of the soil,
and $r_s$ is the average unit weight of soil above the slip surface. $\varphi^b$ is related to the matric suction;
when the matric suction is low, it is close to the internal friction angle(Zhang et al., 2018). $D_s$ is the
thickness of the soil layer above the slip surface. $u_s$ represent the matric suction, which can be
described by the Van Genuchten model(Van Genuchten, 1980):

$$S_e = \frac{\theta - \theta_r}{\theta_s - \theta_r} = \left[\frac{1}{1 + (\alpha_w \times u_s)^n}\right]^m$$
(8)

Where $S_e$ represents the saturation degree, $\theta$ denotes the soil water content of the HSU, $\theta_s$ and $\theta_r$ are
the saturated and residual water content, respectively. The parameters $\alpha_w$, $n$ and $m$ characterize the
shape of the soil–water characteristic curve, with the relationship $n=1-1/m$ .

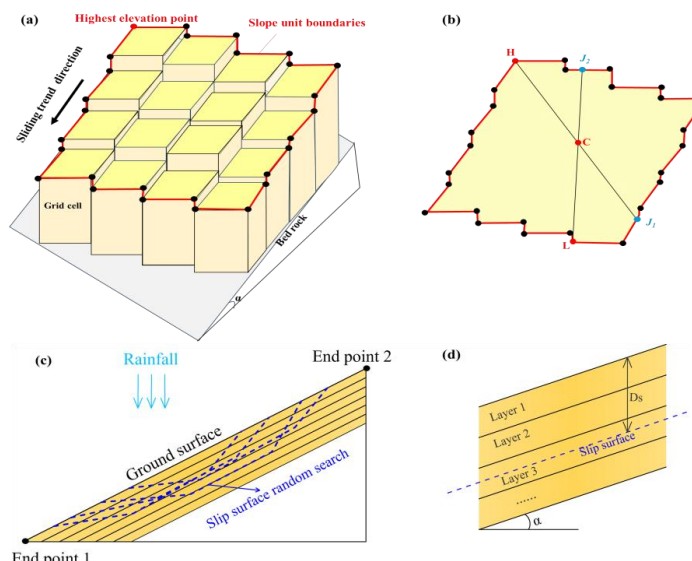

Figure 3 HSU instability probability calculation diagram a. Extraction of HSU boundary points; b. Profile line
extraction of HSU polygon; c. Random search for critical slip surface; d. Enlarged view of the sliding mass for
detailed visualization.



(3) $HSU_{prob}$ calculation
According to the saturated-unsaturated rainfall infiltration theory, the mechanical parameters
of the soil (such as cohesion force $c(kPa)$ and internal friction angle $\varphi(°)$) are significantly affected
by  soil moisture content fluctuations. The variation in soil mechanical parameters during the
process of rainfall infiltration is very complex, and it is generally acknowledged that dry soil prior
to rainfall infiltration exhibits higher mechanical strength (characterized by elevated parameter
values). As rainwater continues to infiltrate, the soil water content gradually increases, leading to a
decreasing trend in mechanical parameters, such as cohesion and internal friction angle.
Consequently, the mechanical parameters of the soil within each HSU are not fixed, but spatial
uncertainty exists to some extent. In this context, employing probabilistic analysis methods to
calculate the instability probability of an HSU is a more reasonable choice. Probability density
functions (such as normal or uniform distributions) are commonly used to describe the uncertainty
of the geotechnical parameters. The normal distribution is considered suitable for small areas or
watersheds where hydrogeological parameters can be collected in detail, whereas a uniform
distribution is more applicable for larger areas, where it is difficult to acquire detailed
hydromechanical parameters(Wang et al., 2021; Wang et al., 2023).
In this study, we utilized a uniform distribution to simulate the uncertainty of the mechanical
parameters within the HSUs. The soil mechanical parameters in the unsaturated state before rainfall
were taken as the upper bound, while those in the fully saturated state were considered the lower
bound, thereby establishing the upper and lower value boundaries for the mechanical parameters
within the HSU, as indicated in Equations (9) and (10):

$$c \in \left[ c_{lower}, \ c_{upper} \right] \tag{9}$$

$$\varphi \in \left[ \varphi_{lower}, \ \varphi_{upper} \right] \tag{10}$$

where $c_{upper}$ and $c_{lower}$ represent the upper and lower bounds of $c(kPa)$, respectively, $\varphi_{upper}$ and
$\varphi_{lower}$ represent the upper and lower bounds of $\varphi(°)$, respectively. The Monte Carlo method was
employed to randomly select the values within these bounds. The instability probability of the HSU
was calculated using Equation (11).
$$HSU_{prob} = \frac{Sum_{Fs<1}}{m} \tag{11}$$

where $m$ represents the number of random selections for the mechanical parameters and $m$



1  is set to 500.

**2.4 The obtainment of key fitting parameters α and β for I-D curves of HSUs**

In this study, an HSU is regarded as unstable when the value of *HSU$_{prob}$* exceeds 50%. Then, the rainfall intensity and duration data with HSU instability under different rainfall scenarios were recorded to obtain the key fitting parameters *α* and *β* for the I-D curves of each HSU, thereby establishing a database of parameters *α* and *β*. The detailed steps are as follows.

(1) Setting the antecedent effective rainfall levels AER_$i(i=1,2,3…n)$

The antecedent effective rainfall(AER) has a significant impact on landslide occurrence. Previous research indicates that in the Three Gorges Reservoir area, the minimum value of AER before landslide occurrence is 0 mm, whereas the maximum value of AER can exceed 170 mm(Wang et al., 2021). Therefore, 20 different levels of AER ranging from 0 to 200 mm were established with intervals of 10 mm.

(2) Design of the combination of rainfall intensity (I) and duration(D)

For each antecedent rainfall level, we categorized rainfall intensity (I) into eight levels to represent the variation from light to heavy rainstorms: 2, 5, 10, 20, 30, 40, 50, and 60 mm/h. The rainfall duration (D) ranged from 1 to 24 h, with intervals of one hour. Consequently, 192 combinations of I and D were generated for each AER level.

(3) Generation of fitting parameters *α*、*β* of the I-D curves

For each combination of rainfall intensity and duration data, the method outlined in Section 2.2 is used to determine the soil water distribution within each HSU, and the corresponding value of HSU$_{prob}$ was computed using the method described in Section 2.3. If the HSU is unstable, the corresponding intensity and duration data can serve as data points for fitting the I-D curves. Subsequently, a power function was utilized to fit these data points to obtain the key fitting parameters α and β of the I-D curve. As presented above, the fitting parameters α and β for the I-D curve of each HSU can be generated, thereby establishing a database for α and β at different AER levels.

**2.5 Warning Mode**

In practical applications, the antecedent effective rainfall(AER), rainfall intensity (I), and duration (D) for each HSU can be computed using Quantitative Precipitation Estimation (QPE) and Quantitative Precipitation Forecasting (QPF) products provided by the meteorological department

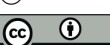



(Wang et al., 2021). Next, we analyzed the relationship between the actual value of AER and the 20
levels of AER documented in the database, thereby determining the level that is closest to the
antecedent effective rainfall data of the HSU. The α and β values corresponding to this level were
retrieved from the database for the following assessments.

5        (1) If $I \geqslant \alpha D^{\beta}$, the data point (I, D) is above the warning curve; thus, the warning should be

released.

7        (2) Conversely, if $I < \alpha D^{\beta}$, it signifies that the data point (I, D) is below the warning curve;

therefore, no warning should be issued.

9        The programming languages Fortran 95 and Python 3.1 were employed to compile the

algorithms outlined in Sections 2.1-2.5, and the overall flowchart of the warning mode is depicted
in Figure 4.

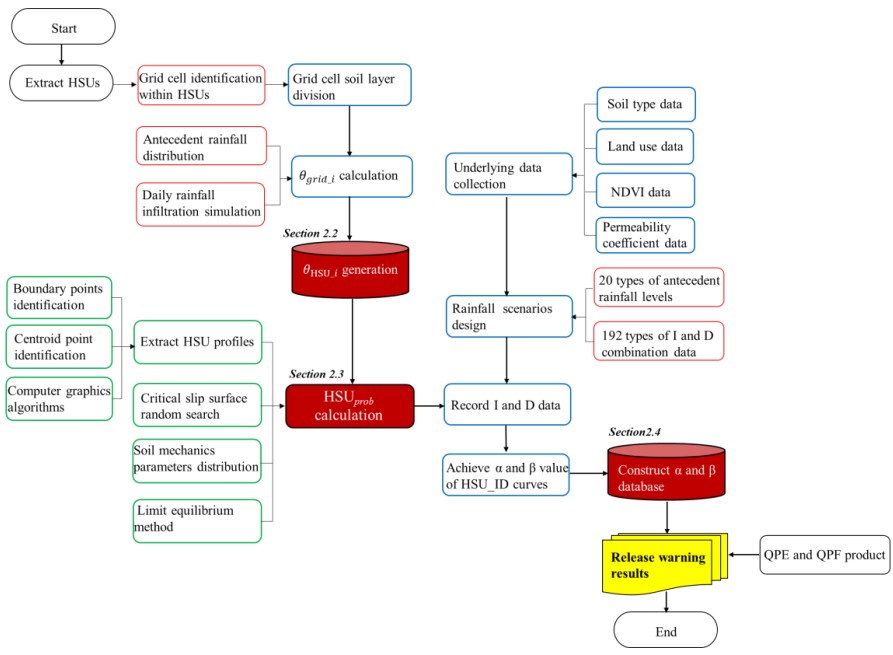

13        Figure 4    The flow chart of the fast warning mode based on parameter α and β database

**3    Study area and data**
**3.1    Study area and slope unit data**

16       Fengjie County is situated in the eastern region of the Three Gorges Reservoir area, with

geographical coordinates ranging from 109°1'17″ " to 109°45'58" East and 30°29'19" to 31°22'33"″
North, covering a total area of 4087 km². The region has a subtropical humid monsoon climate with



an annual average rainfall of 1,500–2,000 mm. The rainy season occurs from May to September,
accounting for 70% of the annual precipitation. The terrain is primarily mountainous and the
Yangtze River flows across the region from west to east. Geological hazards, such as landslides,
debris flows, and collapses, are widely distributed in Fengjie County, with rainfall landslides posing
the most significant threat. Based on the 7m DEM of Fengjie (Figure 5a), the MIA-HSU method
was employed to extract the slope units, resulting in the identification of 17,547 HSUs(Figures 5 b
and c).   Histograms of the slope gradient and area distribution of the HSUs are presented in Figure
5d-e. As shown in Figure 5d, the slope gradients of the HSUs follow a normal distribution, with
85.4% of the slopes falling within the range of 10° to 30°.   Figure 5e illustrates that the average
area of the HSUs is 0.23 km², with 53.9% of the slope units having an area less than 0.25 km².
Because the sliding depth of shallow landslides typically ranges from 2 m to 3 m, the majority of
HSUs can be classified as small-to medium-scale landslides (with volumes under 500,000 m³).

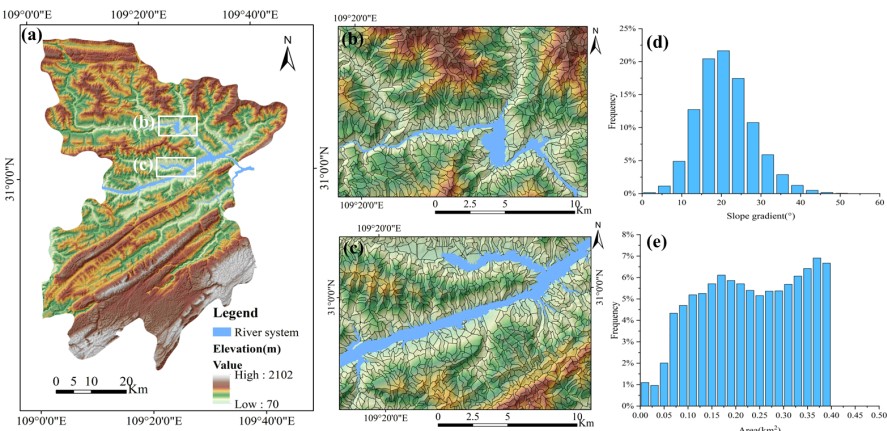

Figure 5 Division of HSUs in Fengjie County a. Fengjie DEM; b and c. Extraction results for selected regions:
Enlarged View; d. Histogram of slope distribution of HSUs; e. Histogram of area distribution of HSUs.
**3.2   Soil mechanical parameter c (kPa) and φ (°) data of HSUs**
The rainfall-triggered shallow landslides within the study area are mainly composed of
quaternary clay and silt, which are classified as fine-grained soils(Wang et al., 2021; Wang et al.,
2023).   Field investigations indicate that the sliding soil is fully or even oversaturated, with some
soil mass transitioning into mudflow during the sliding process. The laboratory moisture content
tests revealed that the soil water content under these conditions approached or exceeded the liquid
limit. To obtain detailed soil mechanical parameters under different moisture states, we conducted



extensive field sampling across Fengjie County, resulting in 312 sampling points, as depicted in
Figure 6f. For each sampling point, direct laboratory shear tests were performed to derive the soil
mechanical parameters c (kPa) and φ (°) at the liquid and plastic limits, respectively. Subsequently,
ArcGIS spatial analysis tools were utilized to generate distribution maps of c (kPa) and φ (°) under
plastic and liquid limit moisture conditions, as shown in Figures 6g-j.

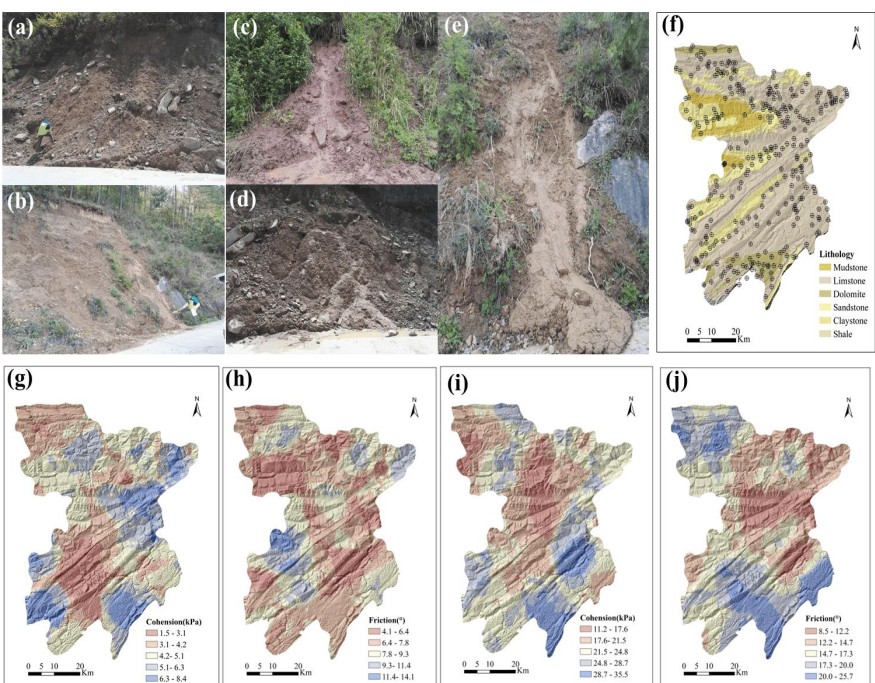

Figure 6 State of Landslide Soil Before and After Rainfall (a. Soil approaching plastic limit moisture content before
rainfall; b. Soil approaching plastic limit moisture content before rainfall; c. Soil in a fluid state after rainfall; d. Soil
in a fluid state after rainfall; e. Fully saturated and liquefied soil after rainfall; f. Soil sampling locations; g. *c* (kPa)
at plastic limit moisture content; h. *φ* (°) at plastic limit moisture content; i. *c* (kPa) at liquid limit moisture content;
j. *φ* (°) at liquid limit moisture content.)
**3.3 Rainfall data**
Rainfall data sources include Quantitative Precipitation Forecasting (QPF) products and
Quantitative Precipitation Estimation (QPE) products. The QPF product obtained from the local
government of Fengjie County is typically utilized to forecast future rainfall at a regional scale,
which can provide rainfall forecast products for the next hour. QPE data are applied to estimate
historical regional rainfall at a regional scale and are essential for determining the antecedent
effective rainfall (AER), which can be computed as follows:



$$AER = \sum_{i=1}^{n} a^n R_i \tag{12}$$

Where AER is the antecedent effective rainfall, *a* is the attenuation coefficient, which is equal
to 0.84, based on the research of the Fengjie count (Wang et al., 2021), *n* is the number of days
before the landslide occurs.
**4 Case Study: Rainfall-induced landslides of 31 August, 2014**
From August 30–31, 2014, Fengjie experienced continuous heavy rainfall, triggering a series
of landslide hazards that resulted in over 30 fatalities and an economic loss of 580 million yuan.
Based on the daily QPE data for August 15-31, the effective precipitation for the 15 days prior to
the landslide hazards is shown in Figure 7. As illustrated in Figure 7, the maximum precipitation
during this period was 179.10 mm, which occurred in the northwestern region of the area. The
hourly QPF data for August 31 are presented in Figure 8 *a-l*. As illustrated in Figure 8 *a-d,* the
rainfall was minimal from 00:00 to 08:00, with a maximum cumulative rainfall of 12.2 mm 08:00.
As shown in Figures 8e-g, rainfall began to increase rapidly at 10:00, reaching a maximum
cumulative precipitation of 92.40 mm by 14:00 in the northwestern region of Fengjie County.
Figures 8 h-l indicate that from 16:00 to 24:00, the cumulative rainfall remained constant,
suggesting that the rainfall process had ceased.

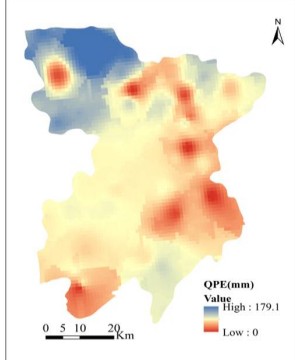

Figure 7 Precipitation Data Processing (Effective Precipitation from August 15 to August 30, 2014)



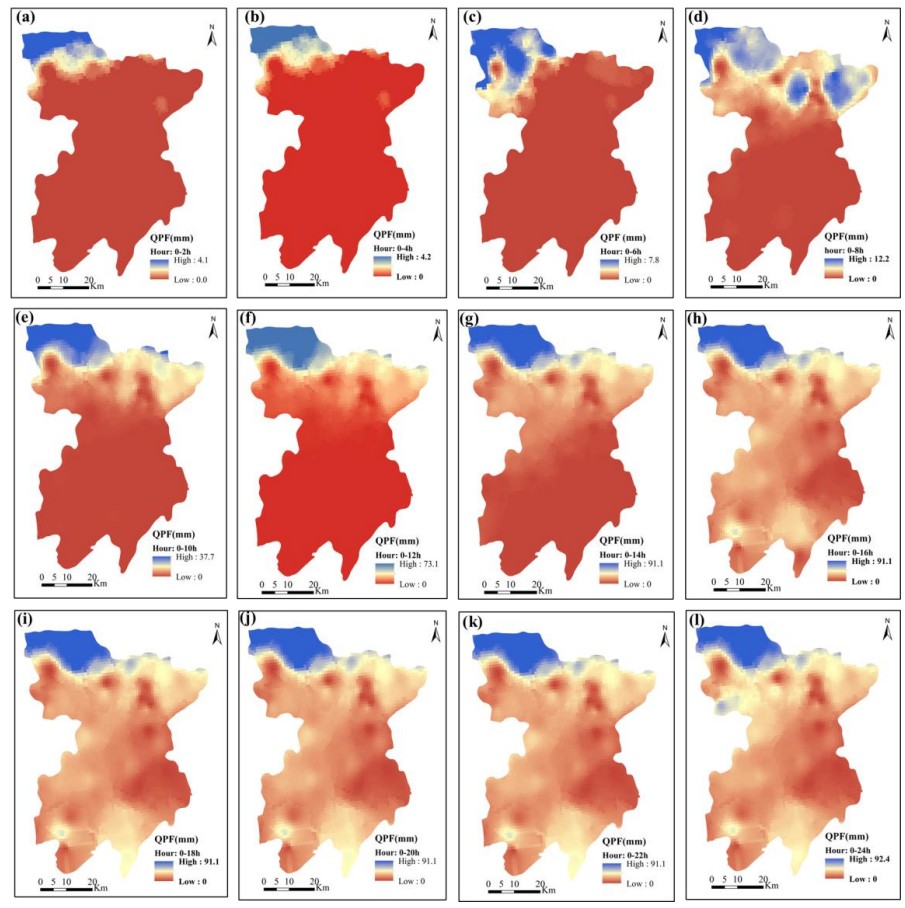

Figure 8 Radar forecast precipitation data for 2014/08/31 (a. 2:00; b. 4:00; c. 6:00; d. 8:00; e. 10:00; f. 12:00; g.
14:00; h. 16:00; i. 18:00; j. 20:00; k. 22:00; l. 24:00)
The Land and Resources Bureau of Fengjie County provided data on landslide points
triggered by rainfall on August 31.    This heavy precipitation triggered 583 landslides, which were
mainly distributed in the northwestern region (as indicated by the red and green solid points in
Figure 9). This study utilized the QPE (Figure 7) and QPF data (Figures 8 *a-l*) as inputs to forecast
landslide hazards for August 31.
The landslide forecast results from 02:00 to 24:00 are shown in Figures 9(a-l). It can be seen
from Figures 8 and 9 that there is a good correlation between the spatial distribution of unstable
HSUs and rainfall characteristics. As presented in Figures 9a-d, at the beginning of the rainfall
process (before 8:00), the majority of the HSUs remained stable owing to the minimal rainfall.
Unstable HSUs began to emerge in the northwestern region starting at 10:00, coinciding with the



rapid increase in rainfall. Additionally, as the rainfall progressed, the number of unstable HSUs
increased swiftly and spread towards the central and southern regions (Figures 9f-g). Notably, many
unstable slope units appeared within several hours after heavy rainfall ceased. Figures 9h-l reveal
that from 16:00 to 24:00, although the heavy rainfall essentially ended, the number of unstable HSUs
continued to rise because of the moisture infiltration of the saturated top soil, reaching a total of
3,987 at 24:00.

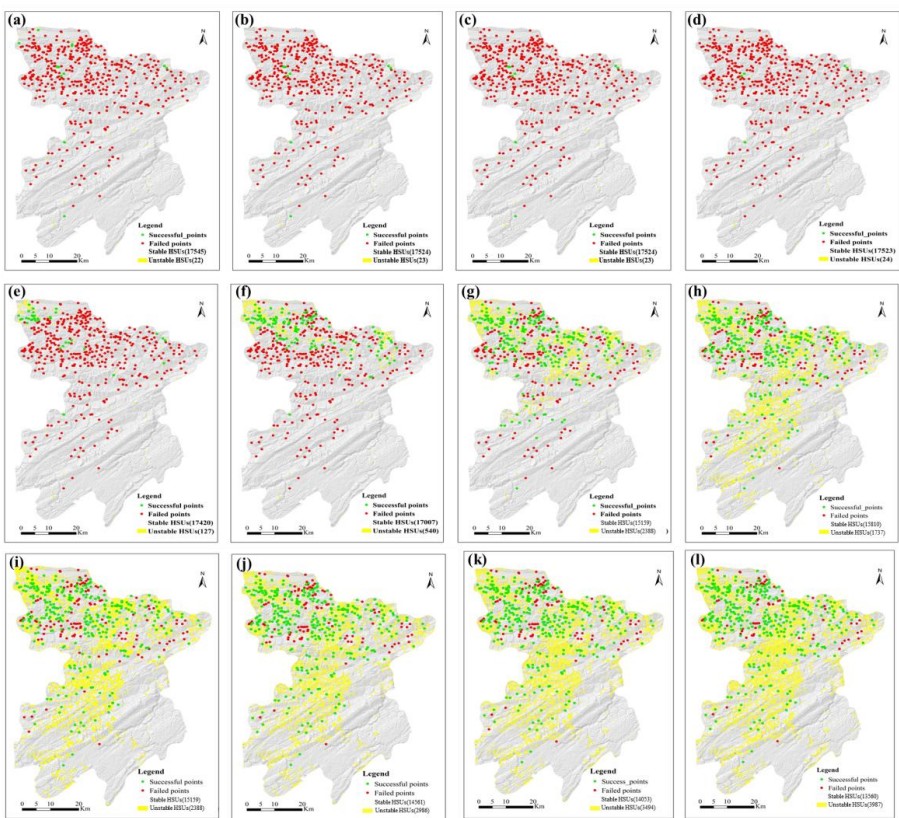

Figure 9 Prediction results at 02:00 to 24:00 (a. 2:00; b. 4:00; c. 6:00; d. 8:00; e. 10:00; f. 12:00; g. 14:00; h. 16:00;
i. 18:00; j. 20:00; k. 22:00; l. 24:00)
This study employs the Receiver Operating Characteristic (ROC) method to analyze the
predictive performance of the HSU(Fawcett, 2006). For physically model-based slope units, the
ROC method describes the following four possible states using a contingency table:
① True Positive (TP): HSU contain landslide points and exhibit instability;
② True Negative (TN): HSU does not contain landslide points and does not exhibit instability;





1 ③False Positive (FP): HSU does not contain landslide points but exhibits instability;

2 ④False Negative (FN): HSU contains landslide points but does not exhibit instability.

3 According to GIS spatial statistics, 583 landslides triggered by rainfall on August 31 were

4 contained within 425 HSUs. In this study, these HSUs are taken as benchmark values for the

5 calculation of TP, TN, FP, and FN, and the missing alarm rate (MAR) and false alarm rate (FPR)

6 can be calculated as follows:

$$MAR = 100\% \times FN / 425 \tag{13}$$

$$FPR = 100\% \times FP / (FP + TN) \tag{14}$$

9 The detailed forecast results for–02-24h are shown in Table 1. As shown in columns 7-8 of

10 Table 1, with increasing rainfall duration, the Missing Alarm Rate (MAR) gradually decreases,

11 while the False Positive Rate (FPR) gradually increases. Taking the result of 24h as an example, the

12 MAR of 24h is 11.8% and the FPR is 21.1%, indicating that the prediction result can satisfy the

13 requirement of early warning practice.

14  Table 1 Analysis of Forecast Results for the 831 Case Study

| Forecasting hour(h) | Unstable HSUs | TP | TN | FP | FN | MAR (%) | FPR (%) |
|---|---|---|---|---|---|---|---|
| 02 | 22 | 7 | 17097 | 25 | 418 | 98.4 | 0.1 |
| 04 | 23 | 3 | 17102 | 20 | 422 | 99.3 | 0.1 |
| 06 | 23 | 3 | 17102 | 20 | 422 | 99.3 | 0.1 |
| 08 | 24 | 2 | 17100 | 22 | 423 | 99.5 | 0.1 |
| 10 | 127 | 27 | 17022 | 100 | 398 | 93.6 | 0.6 |
| 12 | 540 | 116 | 16698 | 424 | 309 | 72.7 | 2.5 |
| 14 | 1370 | 231 | 15983 | 1139 | 194 | 45.6 | 6.7 |
| 16 | 1737 | 289 | 15674 | 1448 | 136 | 32.0 | 8.5 |
| 18 | 2388 | 327 | 15061 | 2061 | 98 | 23.1 | 12.0 |
| 20 | 2986 | 354 | 14490 | 2632 | 71 | 16.7 | 15.4 |
| 22 | 3494 | 364 | 13992 | 3130 | 61 | 14.4 | 18.3 |
| 24 | 3987 | 375 | 13510 | 3612 | 50 | 11.8 | 21.1 |

15 According to the ROC method, the precision and accuracy of the prediction results were

16 calculated as follows:

$$Precision = TPR / (TPR + FPR) \tag{15}$$

$$Accuracy = (TP + TN) / (TP + FN + TN + FP) \tag{16}$$

19 Table 2 provide the calculation results of precision and accuracy at 24h. As shown, the

20 precision of the forecasting results is 80.7%, with an accurancy value of 79.1%, indicating the

21 proposed warning mode has satisfactory comprehensive forecasting performance.



1                           Table 2 Calculation Results of Precision and Accuracy at the 24th Hour

| Forecasting hour(h) | Unstable HSUs | TP | TN | FP | FN | Precision(%) | Accurancy(%) |
|---|---|---|---|---|---|---|---|
| 24 | 3987 | 375 | 13510 | 3612 | 50 | 80.7 | 79.1 |

**5    Discussion**
**5.1 The discussion on the computational efficiency**

4         For emergency warnings during the rainy season, the swift release of warning information is

crucial for local authorities to develop emergency plans and to evacuate residents from landslide-
prone areas. Therefore, local governments not only seek satisfactory accuracy in the warning model
but also require minimal time. To evaluate the computational efficiency of the proposed model, a
standard laptop was utilized to execute the forecast for landslide hazards on August 31. The device
specifications and computation times are presented in Table 3. As shown in Table 3, for the regional
scale covering several thousand square kilometers, the prediction model can rapidly complete real-
time warnings for the next 24 h within 12 min, indicating that its computational efficiency can
satisfy the requirements of emergency warning.

13                           Table 3 Analysis of computational efficiency of the prediction model

| Area (m²) | Number of HSU | CPU | System | Equipment name | Memory | Run time |
|---|---|---|---|---|---|---|
| 4080 | 17547 | Intel(R) Core i7 | Windows 64-bit operating system | ThinkPad P15 Workstation | 16G | <12min |

**5.2 Further Analysis of Prediction Performance**

15        Using the 24-hour prediction results as an example, we randomly selected seven HSUs with

false alarms for further analysis(Table 4). Columns 3–5 of Table 4 present the effective antecedent
rainfall AER of these HSUs, the AER levels assigned by the database, and the relative errors,
respectively. As shown in Column 5, the relative error ranges from 0.7% to 6.3%, indicating that the
20 levels of the AER designed in the database can accurately reflect the effective antecedent rainfall
characteristics of the HSUs. The average rainfall intensity, duration, and cumulative rainfall data at
24:00 are shown in Columns 6–8. As seen in Column 6, the cumulative rainfall for the seven HSUs
ranges from 12 mm to 29.8 mm, with average rainfall intensities range from 0.5 mm/h to 1.25 mm/h,
which can be classified as light to moderate rain type. The instability probability ($HSU_{prob}$) of these
HSUs was calculated to investigate the causes of false positives. As shown in Column 9, among the
seven HSUs with false alarms, five had an instability probability of less than 50%, indicating that



these HSUs did not experience instability during the rainfall process. Therefore, we cautiously
conclude that although the prediction model exhibits preferable operational efficiency, it may
increase the false-positive rate to some extent.
Table 4 The selected HSUs that report false alarms at 24:00

| Number of HSUs | Slope gradient (°) | AER assignment | | | Daily accumulated rainfall (mm) | Duration (h) | Rainfall intensity $I$ (mm/h) | $HSU_{prob}$ |
|---|---|---|---|---|---|---|---|---|
| | | Actual AER (mm) | AER levels assigned by the database (mm) | Relative error | | | | |
| 6172 | 19.4 | 74.6 | 70 | 6.2% | 24.2 | 24 | 1.0 | 0.88 |
| 8561 | 26.5 | 70.9 | 70 | 1.3% | 12.2 | 24 | 0.5 | 0.19 |
| 6066 | 26.7 | 83.1 | 80 | 3.7% | 29.8 | 24 | 1.25 | 0.65 |
| 8535 | 25.9 | 68.6 | 70 | 2.0% | 10.9 | 24 | 0.45 | 0.18 |
| 13108 | 40.3 | 74.1 | 70 | 5.5% | 15.4 | 24 | 0.64 | 0.29 |
| 8297 | 23.4 | 70.5 | 70 | 0.7% | 12.0 | 24 | 0.5 | 0.14 |
| 12966 | 38.3 | 74.7 | 70 | 6.3% | 14.6 | 24 | 0.61 | 0.25 |

To investigate the potential for reducing the false-alarm rate, we selected four HSUs from Table
4 for further analysis. Figures 10 a-d presented the I-D curves and cumulative precipitation
distribution histograms for these HSUs.
For each HSU, the QPF data from 00:00 to 24:00 were discretized into 12 sets of rainfall
intensity and duration data points at 2-hour intervals (represented by black and red solid dots). The
black solid dots positioned below the I-D curve indicate that the HSU is stable at that moment,
whereas the red solid dots located above the curve signify false alarms at the current forecasting
hour. As shown in Figures 10a-d, the red false alarm points for the four HSUs are all situated very
close to the I-D curve, nearly tangent to it. This proximity suggests that slight spatial adjustments
to these points could alter the forecast results. Another important issue is that some of the black
solid dots correspond to a cumulative rainfall of 0 mm, indicating that the rainfall process had not
yet begun. Therefore, it is necessary to adjust the spatial positions of data points I and D based on
the actual initiation time of the rainfall process, thereby facilitating an in-depth investigation of the
causes of the false alarms.



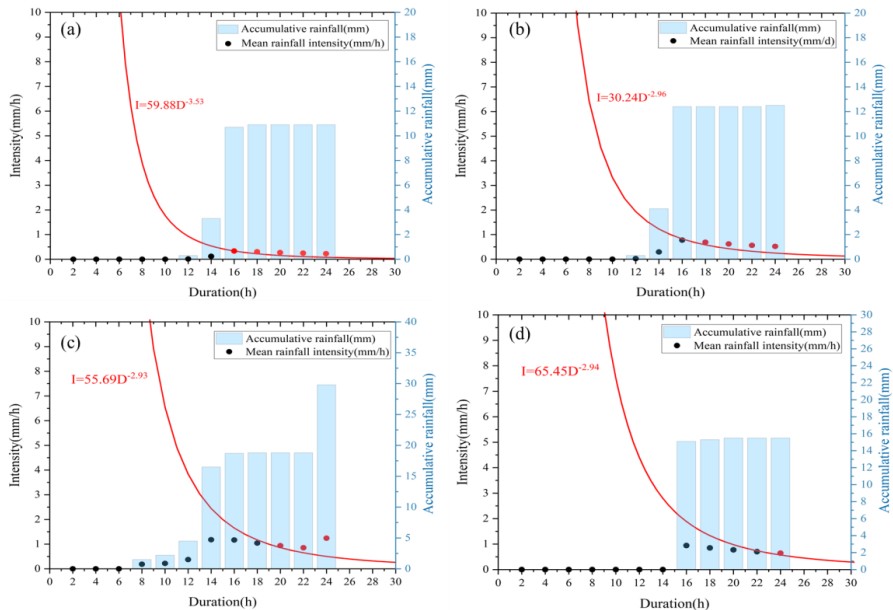

Figure 10 The I-D Curves of HSUs before the adjustment of rainfall process (a. 8535; b. 8561; c. 6066; d. 13108)

In this study, an HSU with number 8535 is taken as an example to illustrate the process of adjusting the spatial positions of data points I and D. As shown in Table 5, the rainfall process for this HSU started at 12:00 and ended at 24:00 with a duration of 12 h. The start time of rainfall was used as the starting point to recalculate the rainfall intensity during the rainfall process, as indicated in the text highlighted with a yellow background in Table 5. The adjusted average rainfall intensity was significantly higher than the values prior to adjustment. This means that the adjustment of the rainfall process led to notable changes in the spatial locations of the data points I and D. As shown in Figure 11a, after updating the positions of data points I and D, the HSU does not exhibit any false alarms. Figures 11b-d present the updated forecast results for the other three HSUs after the adjustment. As shown in Figure 11a-d, following the adjustments, three out of these four HSUs were able to release accurate results. Therefore, we advise that practical warning applications should consider the influence of the difference in rainfall processes of HSUs on the prediction results.

Table 5 Rainfall process adjustment for HSU with number of 8535

| Time | | 2:00 | 4:00 | 6:00 | 8:00 | 10:00 | 12:00 | 14:00 | 16:00 | 18:00 | 20:00 | 22:00 | 24:00 |
|---|---|---|---|---|---|---|---|---|---|---|---|---|---|
| Accumulated rainfall(mm) | | 0 | 0 | 0 | 0 | 0 | 0.3 | 3.3 | 10.7 | 10.9 | 10.9 | 10.9 | 10.9 |
| Before adjustment | I(mm/h) | 0 | 0 | 0 | 0 | 0 | 0 | 0.2 | 0.7 | 0.6 | 0.5 | 0.5 | 0.5 |



| | D (h) | 2 | 4 | 6 | 8 | 10 | 12 | 14 | 16 | 18 | 20 | 22 | 24 |
|---|---|---|---|---|---|---|---|---|---|---|---|---|---|
| After adjustment | I(mm/h) | / | | | | | 0 | 1.6 | 2.7 | 1.8 | 1.3 | 1.1 | 0.9 |
| | D (h) | / | | | | | 0 | 2 | 4 | 6 | 8 | 10 | 12 |

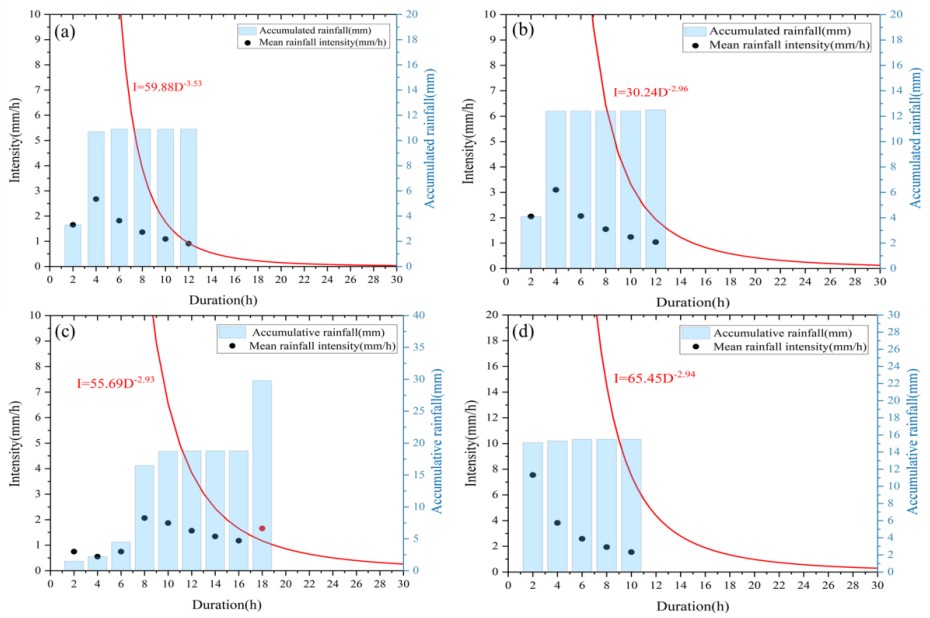

Figure 11 The I-D Curves of HSUs after the adjustment of rainfall process (a. 8535; b. 8561; c. 6066; d. 13108)

## 6 Conclusion

Currently, the operational forecasting of rainfall-induced landslides over regional scales of thousands of square kilometers faces significant challenges. Conventional physical and statistical approaches have shown limitations in terms of achieving satisfactory results. This study utilized HSU as a basis to integrate physical models and rainfall threshold methods for a warning model applicable to large-scale regions. The warning model employs HSU as a prediction unit to improve the clarity of the warning results, physical methods are utilized to develop the warning criteria, thereby reducing the overreliance on historical observational data, and a database of rainfall parameters across different rainfall scenarios is constructed, which enhances the efficiency and applicability of the warning model. The prediction performance was validated through a case study of "8.31" rainfall landslides in Fengjie County. The conclusions are as follows.





(1) A rainfall-triggered landslide warning model was established by integrating HSUs, physical
approaches, and rainfall parameters.   Initially, a grid-based HSU hydrological analysis technique
was established to determine the soil moisture content distribution within the HSUs during different
rainfall hours.   Subsequently, computer graphics algorithms, random search techniques, and
infinite slope models were used to develop a regional-scale HSU stability analysis method. Soil
mechanics parameters at the limit of water content and probability density functions were used to
describe the spatial uncertainty of the soil mechanical parameters within the HSU during rainfall
infiltration, allowing for the calculation of the instability probability of the HSU.   Different rainfall
scenarios were simulated to derive rainfall intensity I and duration D data that can trigger HSU
instability, thereby constructing early warning curves for the rainfall thresholds of the HSU.
(2) A database for the I-D curve fitting parameters $\alpha$ and $\beta$ across various AER levels was
established. This database includes $\alpha$ and $\beta$ data for 17,547 HSUs across 20 AER levels, amounting
to a total of 350,940 records, thus offering substantial data support for rainfall-induced landslide
predictions in Fengjie County. In practical applications, it is sufficient to quickly issue warning
information by assessing the relationship between the values of I and $\alpha D^{\beta}$, thereby reducing the time
required to calculate the safety factors using conventional physical models. The calculation
efficiency test indicates that the warning mode can perform forecasts for thousands of kilometers
within a runtime of less than 12 min, thereby meeting the operational needs for real-time warnings
over large regional scales.
(3) The case study indicates that the distribution trends of unstable HSUs align well with
rainfall characteristics. As the rainfall duration increased, the missing alarm rate (MAR) gradually
decreased, while the false alarm rate (FAR) continued to increase. Taking the 24-hour forecast
results as an example, the missing alarm rate was 11.8%, while the false alarm rate was 21.1%. ROC
analysis revealed that the accuracy of the forecast result at this moment was 80.7%, with a precision
of 79.1%, reflecting satisfactory overall forecasting performance. Further discussion of the false
alarm rate suggests that adjusting the spatial locations of rainfall intensity and duration data points
based on the rainfall characteristics of each HSU may be conducive to reducing false alarm rates.



**Acknowledgements**
The authors would like to acknowledge the Chongqing Meteorological Bureau, China for providing
the QPE and QPF data free of charge. We are also thankful to the Land and Resources Bureau of
Fengjie county for their support with the field investigation.
**Conflict of Interest Statement**
All authors declare that they have no conflicts of interest. We declare that we do not have any
commercial or associative interests that represent a conflict of interest in connection with the
submitted work.
**Author contributions**
**KW:** Conceptualization, Writing – original draft, Supervision, Data curation Funding acquisition;
**LX:** Supervision, Visualization, Writing – original draft; **SX:** Investigation, Data curation,
Validation; **SZ:** Methodology, Resources, Validation; **YJ:** Supervision, Validation, Software; **JZ:**
Investigation, Software; **HG:** Investigation, Visualization; **LZ:** Project administration, Visualization;
**ZW:** Project administration, Writing – review & editing; **FQ:** Writing – review & editing.
**Disclosure statement**
No potential conflict of interest was reported by the author(s).
**Funding**
This work was supported by the [National Natural Science Foundation of China] under Grant
[42301083]; and [Key Scientific Research Project of Higher Education Institutions, Henan Province,
China] under Grant [24A170033]; and [General Project of Henan Province Education Science
Planning, China] under Grant [2025YB0116].
**Data availability statement**
The datasets supporting this study are available from the corresponding author upon reasonable
request.

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
