# Peer review of "Constructing physical-based rainfall landslides prediction model"

_EGUsphere, 2025_

## Referee Comment (RC1)

**[General Comments]**

The manuscript presents a novel approach to landslide forecasting by integrating physical methods, rainfall threshold warning methods, and slope unit analysis. The proposed methodology is computationally efficient and has practical applications in regional-scale early warning systems. It is an interesting study and is well-structured. However, several aspects require clarification, deeper discussion, and refinement to strengthen the paper.

Therefore, the article, at current states, needs to be a medium revision, which may be worth publishing for this journal. The following is my comments for further improving the quality of this manuscript.

**[Major Comments]:**

- (1) The use of meteorological QPE/QPF data is critical to triggering warnings, what are the resolutions of QPE and QPF, and will they have any impact on the forecast results
- (2) In Section 3.2, the author needs to supplement the relevant details of the direct shear test, such as how to determine the dry density and moisture content of the experiment? Was the test conducted under drainage conditions or without drainage conditions?
- (3) How did you ensure the number of iterations or slip surface samples was sufficient to yield stable and representative Fs values across thousands of slope units?
- (4) You claimed that the ROC analysis is based on matching predicted unstable HSUs with 583 observed landslide locations, but the method for spatial matching and thresholding is not clearly explained. For example, how are landslide points assigned to HSUs, Please explain.
- (5) Methodology In some cases, I found that you have used abbreviations without mentioning their full forms for the first time. Please fix it. Check all abbreviations. In some headlines, you use lowercase, and on some, uppercase. For example, the discussion on computational efficiency. But in 5.2. You wrote it with uppercase letters (i.e., Further Analysis of Prediction Performance).
- (6) Page6, line 18, why is the soil layer divided into 10 layers, with each layer having a thickness of 0.2 meters? I suggest including a brief justification for each major assumption, either by citing validation from past studies or noting its limitations.

(7) Some minor mistakes, such as line 3 on page 15, "fengjie count" should be changed to fengjie county; A few references are cited incorrectly. For example, Pradhan, A., Lee, S.-R., Kim, Y.-T. 2018. A shallow slide prediction model combining rainfall threshold warnings and shallow slide susceptibility in Busan, Korea. Landslides, 16: 6. 47-659. doi: <a href="https://doi.org/10.1007/s10346-018-1112-z">https://doi.org/10.1007/s10346-018-1112-z</a>. You may wish to remove them.

---

## Author Comment (AC2)

**Comment 1: I think the manuscript needs more clarity in the novelty and its scientific contribution because by reading the introduction and discussion section I was not very clear what is the main contribution of this manuscript in terms of scientific novelty. While it is evident that it is an interesting topic it would be great to have it explicitly stated what is new compared to existing models and how this advances the state of the art.**

**Authors:** Thanks for this comment. At present, the commonly used rainfall threshold warning methods heavily rely on historical rainfall and landslide inventory content, which limits their applicability in the data-scarce areas. Physical methods do not depend on landslide inventories to establish early warning criteria, but the calculating process for the safety factor usually takes a considerable time amount. To address these issues of both statistical and physical methods, the innovation of this paper lies in integrating physical methods, rainfall threshold methods, and HSUs to establish a new warning mode. We adopted physical methods to establish the I-D early warning curve of each HSU, and then established an early warning database containing 20 antecedent precipitation levels and nearly 200 combinations of rainfall intensity and duration. The specific highlights are as follows:

(1)  The hydrological analysis technique for HSU (homogeneous slope unit) based on grid cells is developed.

(2)  HSU is utilized to integrate physical method and rainfall threshold method for the warning model at regional scale.

(3)  Soil mechanics parameters under the Atterberg limits and probability density functions are utilized to describe the variation of c(kPa) and $\varphi(°)$ within HSU during rainfall infiltration.

(4)  A database of I-D curve fitting parameters $\alpha$ and $\beta$ for each HSU has been established under 20 levels of antecedent precipitation.

(5)  The warning model achieves preferable forecasting performance while ensuring satisfactory computational efficiency.

(6)  Adjusting the spatial locations of I(mm/h) and D(h) data points based on the rainfall process of each HSU is beneficial in reducing the false alarm rates.

**Comment 2:** I think the introduction section does not provide a detailed picture of limitations in existing approaches and how your model addresses them. It would benefit from a detailed literature review. For examples, how your approach addresses issues in PINN or machine learning models etc.

**Authors:** Thanks for this comment. In recent years, some researches have used machine learning methods to conduct regional landslide assessments or predictions, achieving good results. Because machine learning methods rely on the selection and analysis of the sample data, we cautiously believe that machine learning methods still essentially fall within the category of statistical methods. Some promising approaches, such as PINN, are essentially a combination of physical and statistical methods. Considering the length of the article, we did not discuss this in detail in the introduction section. In the revised manuscript, we will add the relevant references.

**Comment 3:** I am a bit skeptical about the slope unit generation process. I think further justification is needed on why you did not use more common approaches for slope unit generation like r.slopeunits? how accurate are your slope units? What are the parameters used to generate them?

**Authors:** Thanks for this comment. At present, there are various methods for extracting slope units, such as the DEM-based hydrological process analysis method, the r.slopeunits method, the

Curvature watershed method, the MIA-HSU method, the Multi-Scale image segmentation method, the SUMak method, etc. The key parameters and extraction results differ among these methods. Taking the r.slopeunits method as an example, this approach ensures the internal slope aspect uniformity of slope units by setting parameters such as area threshold and circular variance. In comparison, the MIA-HSU method uses morphological image techniques to ensure uniform slope gradient and aspect within the HSUs, and controls the number of HSUs by regulating the maximum and minimum area thresholds. In recent years, the MIA-HSU method has been applied in the Jiangjiagou area of Yunnan and the Fengjie area of Chongqing, China.

In this paper, we extracted a total of 17,547 HSUs. As shown in Figure 5, the slope gradients of the HSUs follow a normal distribution, with 85.4% of the slopes falling within the range of 10° to 30°. Figure 5e illustrates that the average area of the HSUs is 0.23 km², with 53.9% of the slope units having an area less than 0.25 km². Because the sliding depth of shallow landslides typically ranges from 2 m to 3 m, the majority of HSUs can be classified as small-to medium-scale landslides (with volumes under 500,000 m³). Therefore, we cautiously believe that the HSU extracted in this paper can accurately reflect the terrain characteristics of rainfall landslide in the Fengjie area. Regarding various extraction methods of slope units, the relevant references are as follows:

**DEM-based hydrological process analysis method:** *Turel M, Frost JD (2011) Delineation of slope profiles from digital elevation models for landslide hazard analysis. Am Soc Civil Eng(224):829–836. https://doi.org/10.1061/41183(418)87*

**r.slopeunits method:** *Alvioli Massimiliano, Guzzetti Fausto, Marchesini Ivan. Parameter-free delineation of slope units and terrain subdivision of Italy[J]. Geomorphology, 2020, 358: 107-124.*

**the Curvature watershed method:** *Yan Ge, Cheng Heqin, Jiang Zeyu, Teng Lizhi, Tang Ming, Shi Tian, Jiang Yuehua, Yang Guoqiang, Zhou Quanping. Recognition of fluvial bank erosion along the main stream of the Yangtze River[J]. Engineering, 2022, 19: 50-61.*

**MIA-HSU method:** *Wang Kai, Zhang Shaojie, Ricardo DelgadoTéllez, Wei Fangqiang. A new slope unit extraction method for regional landslide analysis based on morphological image analysis[J]. Bulletin of Engineering Geology and the Environment, 2019, 78(6): 4139-4151.*

**the Multi-Scale image segmentation method:** *Huang Faming, Tao Siyu, Chang Zhilu, Huang Jinsong, Fan Xuanmei, Jiang Shui Hua Li Wenbin. Efficient and automatic extraction of slope units based on multi-scale segmentation method for landslide assessments[J]. Landslides, 2021, 18(11): 3715-3731.*

**the SUMak method:** *Woodard, J. B., Mirus, B. B., Wood, N. J., Allstadt, K. E., Leshchinsky, B. A., & Crawford, M. M. (2024). Slope Unit Maker (SUMak): an efficient and parameter-free algorithm for delineating slope units to improve landslide modeling. Natural Hazards and Earth System Sciences, 24(1), 1-12.*

For the application of the MIA method, please refer to the following literatures:

*Wang, K., & Zhang, S. 2021. Rainfall-induced landslides assessment in the Fengjie County, Three-Gorge reservoir area, China. Natural Hazards, 108, 1-28. https://doi.org/10.1007/s11069-021-04691-z.*

*Wang, K., Zhang, S., Xie, W.-l., & Guan, H. 2023. Prediction of the instability probability for rainfall induced landslides: the effect of morphological differences in geomorphology within mapping units. Journal of Mountain Science, 20, 1249-1265. https://doi.org/10.1007/s11629-022-7789-4.*

**Comment 4 : I think a sensitivity analysis should also be carried out for modelling and forecasting.**

**Authors:** Thanks for this comment. The main contribution of this manuscript is to establish an I-D curve database for each HSU and then use this database to issue early warning information for each HSU. Because the sensitivity analysis requires repeatedly adjusting parameters to identify which key parameters have the greatest impact on the forecasting results, for this study, each adjustment would mean repeating the database construction process, the workload and time consumption is extremely huge. Therefore, this paper does not conduct sensitivity analysis on the parameters. However, in previous research, the authors carried out sensitivity analysis based on physical models of HSU, we found that slope gradient was the most significant factor influencing the forecast results, followed by soil mechanics parameters and daily rainfall, the rele studies can be referred to:

*Wang, K., Zhang, S., Xie, W.-l., & Guan, H. 2023. Prediction of the instability probability for rainfall induced landslides: the effect of morphological differences in geomorphology within mapping units. Journal of Mountain Science, 20, 1249-1265. https://doi.org/10.1007/s11629-022-7789-4.*

**Comment 5:** Why 500 Monte Carlo iterations are sufficient and consider adding convergence tests? Also why the spatial and random cross validations are not performed?

**Authors:** Thanks for this comment. The authors were sorry because there was currently no unified method regarding the number of iterations of the sliding surface. In this manuscript, the Fengjie area is divided into 17,547 HSUs, the random search times for the sliding surface of each HSU are set to 500 times. If the number of searches continues to increase, the calculation time consumption will increase significantly. We have retrieved relevant literature and cautiously consider that the number of iterations in this paper can be used to search for the position of the critical sliding surface.

Regarding the specific search process, the random search method originated from the random walking method proposed by Greco(1996). The authors have made some improvements based on recent research, specifically including two steps: "node detection loop" and "sliding surface overall detection". Because this content is already relatively mature, detailed technical details can be referred to the following literature:

*Greco, V. 1996. Efficient Monte Carlo Technique for Locating Critical Slip Surface. Journal of Geotechnical Engineering, 122, 517-525. https://doi.org/10.1061/(ASCE)0733-9410(1996)122:7(517).*

*Zhang LY, Zhang JM (2006) Extended algorithm using Monte Carlo techniques for searching general critical slip surface in slope stability analysis. Yantu Gongcheng Xuebao Chin J Geotech Eng 28(7):857–862 (in Chinese)*

*Wang, K., & Zhang, S. 2021. Rainfall-induced landslides assessment in the Fengjie County, Three-Gorge reservoir area, China. Natural Hazards, 108, 1-28. https://doi.org/10.1007/s11069-021-04691-z.*

**Comment 6:** I think while proposing a new model it is very important to include quantitative comparison with other models using performance metrics like ROC, precision, and recall. To show that your model works better and it is needed.

**Authors:** Thanks for this comment. In the commonly used regional landslide forecasting models (such as TRIGRS, SINMAP, etc), the warning results are typically obtained directly from the model equations. However, in this manuscript, the warning results for each HSU are derived from the I-D

database containing 20 different antecedent rainfall levels and nearly 200 combinations of rainfall intensity and duration, rather than directly from the model equations. Therefore, the main contribution of this manuscript is to provide a replicable early warning approach to address the limitations of current statistical and physical methods. If a comparison of forecasting results is to be made, it would require building databases under different rainfall scenarios based on the forecasting model equations(For example the infinite slope model equation in the TRIGRS model). We cautiously believe that the amount of work involved would be enormous, and we will conduct comparative analyses of different model databases in future studies.

Regarding the evaluation metrics, we used the missing alarm rate, false alarm rates, precision, and accuracy to evaluate the forecasting results. These quantitative indicators are also derived from the ROC method, as referenced in the literature:

*Fawcett, T. 2006. Introduction to ROC analysis. Pattern Recognition Letters, 27, 861-874. https://doi.org/10.1016/j.patrec.2005.10.010.*

**Comment 7:** I think you should use metric such as F1 Score and ROC/AUC instead of accuracy or just precision to show overall predictive capability of this model.

**Authors:** Thanks for this comment. As mentioned in the previous reply, the quantitative indicators such as missing alarm rate, false alarm rates, precision, and accuracy from the ROC method are used to evaluate the forecast results. During the process of writing this manuscript, we also attempted to plot the ROC curve. Because in this manuscript, whether a landslide occurs depends on the relationship between the rainfall parameters (antecedent effective precipitation, rainfall intensity and duration) and the position of the warning curves in the database, this means that for each HSU, there are only two possible warning outcomes: a landslide occurs or does not occur. Therefore, our warning results are not continuous probability values between 0 and 1, and the conventional method for plotting an ROC curve cannot be applied. While there are some data processing methods currently available to address this issue, the authors remain cautious and consider it more reasonable to keep the data in its original state and to use other quantitative indicators derived from the ROC method for evaluation.

**Comment 8:** Improve Figures and Visualization: Ensure axes labels, units, and legends are clear; consider adding color scales and zoomed views. For example in Figure 7 I cannot read anything in Legend or scale bar.

**Authors:** Thanks for this comment. In the revised version, we will improve the figures and Visualization as required. Regarding Figure 7, it presents the effective precipitation data of the 15 days before the landslide. As can be seen from Figure 7, the northwest region had the highest antecedent effective precipitation (reaching up to 179mm), which is also the area with the most landslides events.

**Comment 9:** I think it is important to add a subsection on limitations and future work, including transferability and data-scarce environments.

**Authors:** Thanks for this comment. We will add some content on the limitations and future work in the modified manuscript.